# Analysis of TiO_2_ Nanolubricant Influence in Micro Deep Drawing of Stainless Steel SUS301

**DOI:** 10.3390/ma16062196

**Published:** 2023-03-09

**Authors:** Di Pan, Guangqing Zhang, Fanghui Jia, Lianjie Li, Tao Zhang, Yao Lu, Hui Wu, Ming Yang, Zhengyi Jiang

**Affiliations:** 1School of Mechanical, Materials, Mechatronic and Biomedical Engineering, University of Wollongong, Northfields Ave., Wollongong, NSW 2522, Australia; 2Graduate School of System Design, Tokyo Metropolitan University, Tokyo 191-0065, Japan; yang@tmu.ac.jp

**Keywords:** micro deep drawing, austenitic stainless steel SUS301, TiO_2_ nanolubricant, wrinkling, lubricant pockets theory

## Abstract

To improve the quality of products produced from microforming, various nanolubricants have been applied in the field of micromanufacturing in recent years. In this paper, the effects of glycerol-based lubricant containing TiO_2_ NPs (NPs) on micro deep drawing (MDD) of austenitic stainless steel (ASS) SUS301 were studied, and the lubrication mechanism involved was discussed. The MDD experiments were conducted with the SUS301 foils under dry, 1, 2, and 4 wt% TiO_2_ NP lubrication conditions. The results show that the use of the TiO_2_ nanolubricants can significantly improve the quality of the drawn cups in terms of decreased wrinkling and surface roughness. Besides, the concentration of TiO_2_ NPs influences lubricity by reducing friction during the MDD process. The peak drawing force is the lowest when 2 wt% nanolubricant is applied, which drops to 72.54 N from 77.38 N under dry conditions. The micro cup drawn under 2 wt% TiO_2_ nanolubricant has the best quality among those obtained under all the lubrication conditions. The lubrication mechanisms are derived from the mending effects of TiO_2_ NPs and the formation of thin lubricant films associated with the open lubricant pockets (OLPs) and close lubricant pocket (CLPs) theory in the MDD. The CLPs function as reservoirs that retain lubricants to counteract the load pressure, whereas the OLPs lead to lubricant leakage due to the higher flow resistance. It was found that the lubricant film and NPs are insufficient at a low concentration (1 wt%), while the lubrication performance can be enhanced with increased NP concentration. However, there exist apparent agglomerations on the surface of the produced micro cup when using 4 wt% nanolubricant, which greatly deteriorates the lubricant performance in the MDD process. It is concluded that the lubricant containing 2 wt% TiO_2_ NPs demonstrates the best lubrication performance during the MDD of ASS SUS301.

## 1. Introduction

Micro deep drawing (MDD) is a fundamental forming method used to produce thin-walled, hollow, box or cup-like metal products at the micro scale [1,2,3]. Recent years have witnessed a growing interest in MDD as a promising micro-manufacturing technology due to its mass production potential, minor operator requirement, and low tool cost [4,5,6,7]. 

Friction in MDD can significantly influence the profile accuracy, height derivation, and surface roughness of the micro cup [4,8,9,10,11,12]. Therefore, reducing friction is of paramount importance in MDD. Traditional liquid lubricants can effectively reduce friction in many macro-scale metal forming processes. However, they cannot be directly applied in the MDD process because the lubricant film is hard to form and maintain under high contact pressure at the micro scale [13,14,15,16]. To address this problem, adding nanoparticles (NPs) into the base lubricant has been identified as one of the best solutions [9,16,17,18]. To further examine the potential of utilising NPs for this purpose, Wu et al. [19] investigated the tribological behaviour of the lubricant with TiO_2_ NPs and found that using the nanolubricant can effectively alleviate the overflow and reduce the friction in the steel-to-steel sliding process. As a result, subsequent research has been dedicated to exploring the correlation between the performance of the nanolubricant and the specific characteristics of the added NPs [20,21,22,23]. Furthermore, other researchers [24,25,26,27,28,29,30,31] have incorporated TiO_2_, SiO_2_, CuS, Al_2_O_3_, and MoS_2_ NPs into liquid lubricants for application in metal rolling. The results showed that the shape and size of added NPs could significantly influence the dispersion stability and performance of the nanolubricant. For example, TiO_2_ NPs with spherical shapes can well disperse in water and, therefore, effectively reduce the friction between mating surfaces [32], and smaller particles exhibit a higher surface area-to-volume ratio, leading to reduced friction and wear [19]. Moreover, the impact of NP size on dispersion stability is crucial in optimising lubricant performance, as demonstrated by previous research showing that certain size ranges can result in improved dispersion stability [33,34]. For instance, P25 TiO_2_ NPs, which consist of both anatase and rutile phases and have an appropriate particle size, have been found to reduce friction between the tool and the workpiece [35,36].

To optimise the lubrication performance of liquid lubricants with TiO_2_ NPs, a comprehensive understanding of their lubrication mechanism is essential. To this end, Wu et al. [37,38] conducted the hot rolling tests under pure water and water-based TiO_2_ nanolubrication conditions, which revealed that the mending and ball-bearing effects of TiO_2_ NPs are the primary causes of reduced friction in the contact zone. Moreover, Ma et al. [39] investigated different fractions of TiO_2_ NPs in the micro rolling and found that a low fraction of TiO_2_ NPs could cause insufficient lubrication. Conversely, excessive TiO_2_ NPs could lead to apparent agglomeration at the contact regions, ultimately deteriorating the surface quality [34,40,41]. Hence, it is crucial to maintain an appropriate concentration of NPs for optimal lubrication performance.

To date, TiO_2_ nanolubricants have been widely used in many engineering applications, such as metal machining, the automotive industry, and the aerospace industry. However, limited research has been conducted on the application of the TiO_2_ nanolubricants in the MDD of austenitic stainless steel (ASS) SUS301, let alone the combination with open lubricant pockets (OLPs) and close lubricant pocket (CLPs) theory. In addition, the impact of NP concentration on lubrication performance during MDD remains poorly understood [42]. To fill in this research gap, MDD of ASS SUS301 was conducted in this study using dry condition and glycerol-based TiO_2_ nanolubricants with varying concentrations of 1, 2, and 4 wt%. The resulting profile and surface quality of the micro cups were then analysed, and the role of TiO_2_ NPs in the MDD of SUS301 was examined in consideration of the OLPs and CLPs theory.

## 2. Materials and Methods

SUS301 foils with a 5 mm × 5 mm (length × width) and thickness of 40 ± 2 µm were used in this study. The chemical compositions of the foil are listed in Table 1. In order to enhance the crystallinity and eliminate the residual stress of the SUS301 foils, they were subjected to annealing in an argon-filled KTL tube furnace at 980 °C for 2 min [43,44]. After the annealing process, micro tensile tests were conducted on 5 occasions to acquire the average mechanical properties of the foils. The precision gearbox used in the tensile tests was the NN60 series from Cuken Company, which featured a deceleration ratio of 100:1. Micro tensile samples with a thickness of 40 μm, a width of 3 mm, and a length of 15 mm were prepared in accordance with the ISO 12086-2:1955 standard. The tensile speed was 0.05 mm/s to achieve a quasi-static condition. As shown in Figure 1, the stress-strain curve obtained from the micro tensile test is presented. The average yield strength of the annealed foils, as determined from the curve, was found to be 387 ± 10 MPa. This value serves as an important indicator of the material’s resistance to deformation and plastic flow.

The MDD experiments were conducted in the press machine DT-3AW, as shown in Figure 2. There are two crucial parts in the MDD system, including the press machine and die sets. The MDD tests were performed in the die set containing the upper and the lower dies, and the drawing speed was set as 0.1 mm/s. The upper die comprises a micro punch, force sensor, and blank holder; the lower die includes the cavity. Table 2 summarises the geometrical dimensions of the die set. 

Different weight concentrations (1, 2, and 4 wt%) of glycerol-based TiO_2_ NP lubricants were prepared to be utilised in the MDD. The TiO_2_ nano-additive glycerol-based lubricants were prepared by the following method: First, a specific amount of pure TiO_2_ NPs (P25 sourced from Sigma-Aldrich™ (St. Louis, MO, USA)) with an approximate diameter of 20 nm) was weighed and added to balanced deionised water. The mixture was then stirred mechanically. Next, 0.4 wt% of polyethyleneimine (PEI) was gradually added as a dispersing agent, and the mixture was centrifuged at 2000 rpm for 10 min to prepare a stable suspension. PEI, a cationic polymer, acts as a surfactant for the TiO_2_ NPs, thereby improving the dispersion of the NPs. Subsequently, 80 wt% of glycerol was added drop by drop to the solution. The suspension underwent mechanical stirring at a speed of 2000 rpm for 10 min, followed by ultrasonication for an additional 10 min to ensure complete disintegration of any remaining clumps. The weight of configuring lubricant is 50 g, and the specific chemical compositions of the various lubrication conditions can be found in Table 3. To determine the phase of TiO_2_ NPs, an X-ray diffraction (XRD) analysis was performed on a sample that was prepared by mixing TiO_2_ NPs with ethanol and depositing them onto a glass substrate. The process involved dipping the mixture onto the substrate and then allowing the ethanol to evaporate, thereby leaving the NPs deposited on the glass surface. The XRD analysis was performed using a Philips PW1730 conventional diffractometer (Royal Philips, Amsterdam, The Netherlands) equipped with Cu-Kα radiation. The measurement parameters were optimised to ensure accurate results, with a 2θ range set between 0–70°, a step size of 0.02°, and a scan speed of 1.5° per minute. The entire analysis was performed under controlled room temperature conditions. In addition, the JEM-ARM200F transmission electron microscope (TEM)from JEOL (Tokyo, Japan) combined with energy dispersive spectroscopy (EDS) was used to obtain the distribution and size patterns of the NPs at different concentrations. The stability of glycerol-based nanolubricants was evaluated using a sedimentation method, which allowed for direct observation of NP sedimentation. The results of dispersion stability were obtained through photo capture within a 120-h period. The viscosities of these lubricants were measured by rheometer MCR 301 (Anton Paar, Sydney, NSW, Australia) at room temperature. To ensure the homogeneity and dispersion of NPs, the lubricants were placed in an ultrasonic bath for 20 min before the MDD. As illustrated in Figure 3, approx. 0.1 mL of the nanolubricant was introduced into the die cavity prior to performing the MDD tests. By doing this, the lubricant adhered to the die cavity, forming a lubricant film which carried the TiO_2_ NPs. 

In this study, the force data collected from the KYOWA load cell LMA-A-200 N (Niza, SP, Japan), which had a capacity of up to 200 N and an accuracy of 0.01 N, were analysed. After the MDD tests, the profiles, morphologies, and EDS mappings of the micro cups were observed by a VK-X100 3D laser scanning microscope from Keyence (Kallang, Singapore) and a JSM-7001F field emission scanning electron microscope (SEM) from JEOL (Tokyo, Japan). To obtain reliable results, the MDD tests were repeated five times under the proposed lubrication conditions.

## 3. Results and Discussion

### 3.1. Characterisation of TiO_2_ Nanolubricant

Figure 4 presents the XRD patterns of the TiO_2_ NPs being used in this study. The phases of these NPs can be determined as a typical P25 TiO_2_, which contains 25 wt% rutile and 75 wt% anatase, referring to the XRD standard atlas as (JCPDS Nos. 21-1272 for Anatase-type and 21-1276 for Rutile-type). Table 4 lists the viscosity values of the lubricants at a shear rate of 1000/s, which showed that the inclusion of TiO_2_ NPs in glycerol increased viscosity from 0.723 ± 0.03 Pa·s to 0.749 ± 0.04 Pa·s, reflecting a 13% increase when the NP concentration increased from 1 wt% to 2 wt%. A slight 4% rise in viscosity was noted when the NP concentration was further increased to 4 wt%. The 2 wt% lubricant had the smallest error bar, indicating its most stable viscosity among the various lubricants. Figure 5 presents TEM images of TiO_2_ NPs at concentrations of 1 wt%, 2 wt%, and 4 wt% immediately after preparation. The results indicate that the NPs maintain their size and dispersion throughout the observation period under various conditions, demonstrating the long-term stability of the system. It is noteworthy that most of the TiO_2_ NPs were found to be uniformly dispersed with an average diameter of around 30 nm. However, at a concentration of 4 wt%, a clear tendency towards agglomeration was observed. 

Figure 6 displays the sedimentation of TiO_2_ NPs in various concentrations over different periods of time. The results indicate that the as-synthesised suspensions exhibit exceptional stability, regardless of composition. After a prolonged period of 120 h, the TiO_2_ NPs remained largely undisturbed, demonstrating the outstanding dispersion stability of the glycerol-based nanolubricants.

### 3.2. Effect of TiO_2_ NP Concentration on Deep Drawing Force

Figure 7a shows the force versus displacement curves under different lubrication conditions. As shown in this figure, the variation trend of drawing force is almost the same under four lubrication conditions. The drawing force rises slightly at the beginning of the drawing process. With the downward movement of the punch, the drawing force increases significantly to the peak value, then decreases to the end. The 4 wt% curve in Figure 7a is distinguished by an earlier peak value compared to the dry, 1, 2, and 4% curves. This may be due to the increased concentration of TiO_2_ nanolubricant, which allows for the blank to slip into the die cavity more quickly. This results in an earlier onset of peak force during the material deformation process. It is notable that this strain in the initial stage may have a significant impact on the quality of the micro cup mouth. It is generally agreed that the contact area between the blank and the cavity keeps increasing during the drawing process, which enhances the proportion of friction. Alternatively, it could simply mean that the peak drawing forces could be compared to reflect the performance of the lubricants in reducing friction. As discussed, the lower peak drawing force indicates more reduction of friction. It can be seen from Figure 7b that the peak drawing forces obtained under dry, 1, 2, and 4 wt% TiO_2_ lubrication conditions are 77.38 ± 2.4, 72.54 ± 3.2, 70.32 ± 1.5, and 71.79 ± 1.2 N, respectively. These results are similar to those obtained in the previous study [42], showing that the peak drawing force is lower under the lubrication case than that under dry conditions. This means TiO_2_ nanolubricant is efficient in reducing friction. It should be noted that the reduction of the peak drawing force is the largest, namely 7.06 N (9.1%), when using the 2 wt% nanolubricant. However, the effect of the nanolubricant on the last drawing force should also be considered, as depicted in Figure 7b. The last drawing forces are 17.13 ± 3.1, 11.63 ± 2.2, 10.86 ± 1.5, and 13.91 ± 0.8 N under dry, 1, 2, and 4 wt% TiO_2_ lubricants, respectively. The last drawing force, which is the force required to complete the micro deep drawing process and produce the final part, is a crucial parameter in the micro deep drawing process. It not only impacts the quality and consistency of the final parts but also the energy consumption and process efficiency. As the final drawing force increases, the total deformation of the material also increases, along with a decline in the surface finish and dimensional accuracy of the final parts. In addition, the final drawing force is also related to energy consumption and process efficiency. A lower final drawing force leads to less energy consumption, which can contribute to a sustainable manufacturing process. Our study found that the 2 wt% TiO_2_ nanolubricant is the most effective in reducing the largest and final drawing force, making it a potentially efficient lubricant for use in the micro deep drawing process.

### 3.3. Effect of TiO_2_ Nanolubricant on the Quality of Cups

Figure 8 presents the mouth view of micro cups drawn under different lubrication conditions. It is notable that the micro cups drawn under the nanolubricants have a smaller number of wrinkling points than those drawn under dry conditions. Besides, the micro cup drawn under 2 wt% TiO_2_ nanolubricant exhibits the least wrinkles. The wrinkling occurs during the MDD due to the inhomogeneous formation along the rim of the blank. In addition, the wrinkling might rely on the energy consumption along the blank rim, where the primary formation is circumferential compression; the secondary formation contains bending and thickness direction blank holding. A difference between the wrinkling of the micro cups can be attributed to the performance of the lubricants. When more energy was consumed in the primary formation than that in the secondary formation, obvious wrinkling could be formed. The TiO_2_ lubricants could help reduce friction, allowing the blank to slide more easily into the die cavity. The energy consumption of the compression could be minimised with less contact time between the blank and blank holder, thus reducing the wrinkling. The results demonstrated that the wrinkling depends on the concentration of the TiO_2_ nanolubricant. According to Section 3.1, the 2 wt% TiO_2_ nanolubricant has the best performance in reducing friction among these lubrication conditions. Therefore, the cup drawn under 2 wt% TiO_2_ nanolubricant exhibits the least wrinkles.

Figure 9a shows the area for the measurement of surface quality. This area is in the middle of the side of the micro cup where the friction influences the surface quality significantly [4]. Therefore, measuring the roughness of this area could explore the effect of the lubricant on the surface quality. Figure 9b–e display the surface view of the micro cups drawn under dry, 1, 2, and 4 wt% TiO_2_ lubrication conditions, respectively. It can be observed that apparent scratches exist on the surface of the micro cup. To evaluate the surface quality of the micro cup statically, the surface roughness *R_a_* could be measured. Prior to the MDD, the surface roughness of the material was 0.34 ± 0.1 µm. Figure 10 shows the average *R_a_* of the micro cup surface under different lubrication conditions. From this figure, the *R_a_* value is 0.65 ± 0.15 µm under dry conditions, and this value decreases when using the TiO_2_ nanolubricants. Besides, the *R_a_* values are 0.42 ± 0.1, 0.35 ± 0.08, and 0.45 ± 0.12 µm under 1, 2, and 4 wt% TiO_2_ nanolubricants, respectively. The results show that the application of nanolubricant could help improve the surface quality of the drawn micro cup. It is notable that the *R_a_* value is the lowest under 2 wt% nanolubricant among all the lubricants being used, which indicates that the micro cup drawn under 2 wt% lubricant has the smoothest surface. During the drawing process, the surface of the micro cup becomes rough due to the friction [45]. The nanolubricants are efficient in reducing friction, and 2 wt% TiO_2_ nanolubricant is the most efficient in reducing the friction. These findings support the notion that using the 2 wt% TiO_2_ lubricant is the most efficient in improving the surface quality of the drawn cup.

### 3.4. Lubrication Mechanism

Figure 11a–c shows the SEM images and EDS mappings of the micro cups drawn under 1, 2, and 4 wt% TiO_2_ nanolubricant, respectively. The figure demonstrates that the TiO_2_ NPs remain on the surface of the micro cup. Besides, the amount of the remaining TiO_2_ NPs increases significantly with a higher concentration of TiO_2_ NPs. When observing the morphologies of the TiO_2_ NPs with different concentrations, it is found that TiO_2_ NPs distribute scattered under 1 wt% TiO_2_ nanolubricant. Besides, the NPs agglomerate more obviously under a denser nanolubricant. The amount of TiO_2_ NPs clusters increases significantly when increasing the concentration of the TiO_2_ nanolubricant from 2 wt% to 4 wt%. The lubricant formed the lubricant film carrying the TiO_2_ NPs on the die cavity surface. It is notable that the attached lubricating film can avoid direct contact between the blank and the die cavity, which then decreases the friction. According to Wilson and Murch [46], the thickness of the lubricating film is related to the dynamic viscosity. The film could be insufficient to spread on the surface if the viscosity is too low. Therefore, increasing the concentration of TiO_2_ NPs could promise a sufficient film. Meanwhile, the TiO_2_ NPs in the lubricant film act as ball bearing that can convert the motion between contact pairs from sliding to rotating, further reducing the friction [47,48]. Additionally, the TiO_2_ NPs can be pressed into surface defects, mending the uneven surface and further enhancing the performance of the lubricants [49]. The result shows that the TiO_2_ NP clusters exist obviously under 4 wt% TiO_2_ nanolubricant. It has been reported that too many NPs existing in a small space could incorporate carrying fluid on the surface of each NP into an effective volume of solid, which, in turn, increases the possibility of agglomeration [19].

Figure 12 shows the distribution of TiO_2_ NPs near the rim of the micro cup drawn under 1, 2, and 4 wt% TiO_2_ nanolubricants. The TiO_2_ clusters occur obviously near the rim, and the amount of these clusters increases significantly with the higher concentration of TiO_2_ NPs. OLPs and CLPs theory could be considered when using the nanolubricant in microforming [11,42,45]. Figure 13 illustrates the formation of a lubricant film at the edge of the sample based on the OLPs and CLPs theories. It is notable that the CLPs area retains the lubricant that could counteract the load pressure. The surface flattens during the forming process. Thus the actual contact area increases. Therefore, the lubricant in the original direct-contact surface and CLPs move to the newly added direct-contact surface and form a lubricating film. The OLPs are in the area adjacent to the surface edge, resulting in the leakage of the lubricant. The lubricant becomes more dense and viscous with a higher concentration, thus increasing the flow resistance that can cause more static behaviour to avoid overflow and drain from the interface. The lubricant trapped within the surface roughness valleys is able to withstand the contact pressure, while the NPs aid in the storage of the lubricant. The contact pressure has a significant impact on friction and, thus, the overall forming load [42,48]. However, at a concentration of 4 wt%, the nanolubricant was found to clump together easily. The agglomeration of the NPs can impede the continuous supply of fine NPs to the contact surface for lubrication, thereby reducing the lubricity. Ultimately, a 2 wt% concentration of TiO_2_ nanolubricant was found to provide an adequate lubricant film while minimising clumping, resulting in the best performance in reducing friction.

## 4. Conclusions

This work presents experimental studies on the micro deep drawing (MDD) of austenitic stainless steel (ASS) SUS301 foils under different lubrication conditions, including dry conditions and glycerol-based lubricants with 1, 2, and 4 wt% TiO_2_ NPs. This aims to clearly understand the lubrication mechanisms involved in MDD and explore an optimal lubricant for the MDD of ASS SUS301. The following conclusions were drawn from this work:The TiO_2_ NPs were well dispersed in the developed glycerol-based nanolubricants. The viscosity of the nanolubricant increased with the increased TiO_2_ NPs concentration, from 0.749 Pa∙s at 1 wt% to 0.872 Pa∙s at 4 wt%.The greatest peak and last drawing forces were observed under dry conditions, and these forces were reduced by 9.1% and 36.6%, respectively, when 2 wt% TiO_2_ nanolubricant was applied.The surface of the micro cup exhibited the least wrinkling and the smoothest surface when lubricated with 2 wt% TiO_2_ nanolubricant.The OLP and CLP theories were proposed to interpret the lubrication mechanism involved in MDD. The CLP theory suggests that the nanolubricant can move to the newly formed surface and further reduce friction, while the OLP theory suggests that the addition of TiO_2_ NPs to the nanolubricant can prevent lubricant leakage. However, when the concentration of TiO_2_ NPs in the nanolubricant is increased to 4 wt%, agglomeration of the NPs can occur, which may impede the continuous supply of fine NPs to the contact surface for lubrication.

## Figures and Tables

**Figure 1 materials-16-02196-f001:**
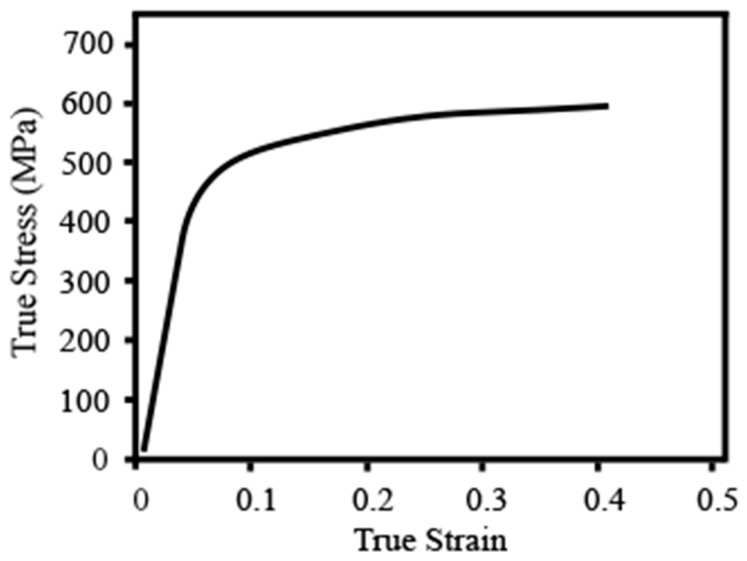
Stress-strain curve of annealed SUS301 foils from micro tensile tests.

**Figure 2 materials-16-02196-f002:**
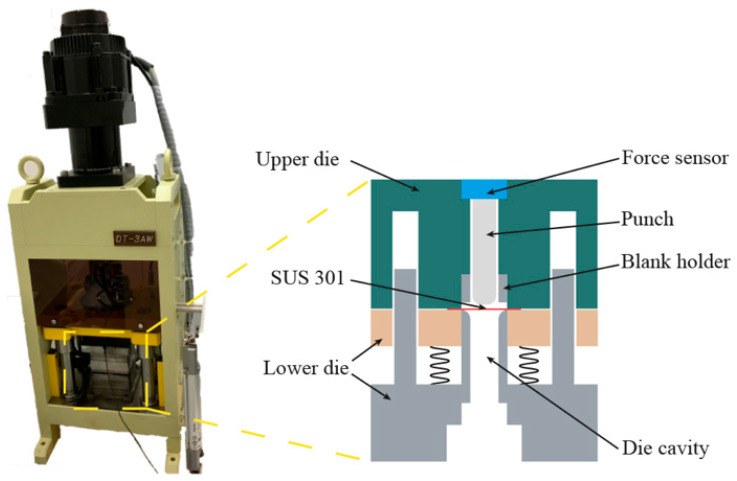
MDD apparatus and its schematic illustration.

**Figure 3 materials-16-02196-f003:**
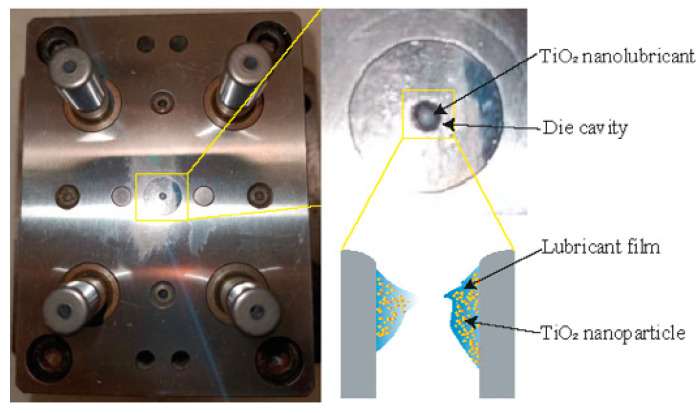
Schematic diagrams of the deep drawing configuration under TiO_2_ NP lubrication condition.

**Figure 4 materials-16-02196-f004:**
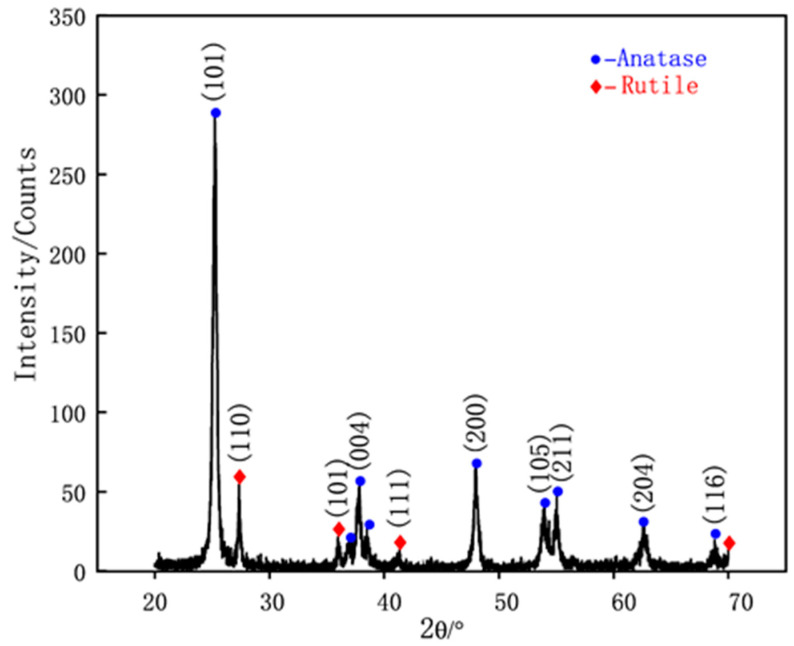
XRD pattern of TiO_2_ NPs.

**Figure 5 materials-16-02196-f005:**
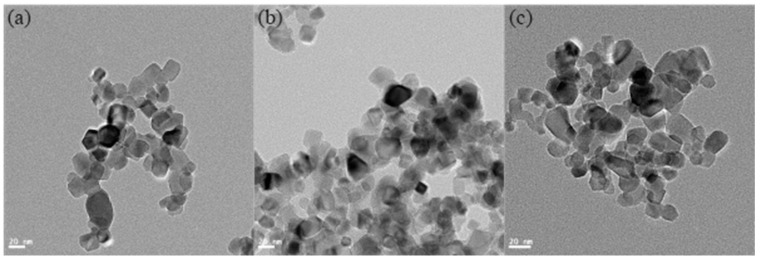
TEM images of TiO_2_ NPs dried from different weight concentrations lubricants: (**a**) 1 wt%; (**b**) 2 wt%; and (**c**) 4 wt%.

**Figure 6 materials-16-02196-f006:**
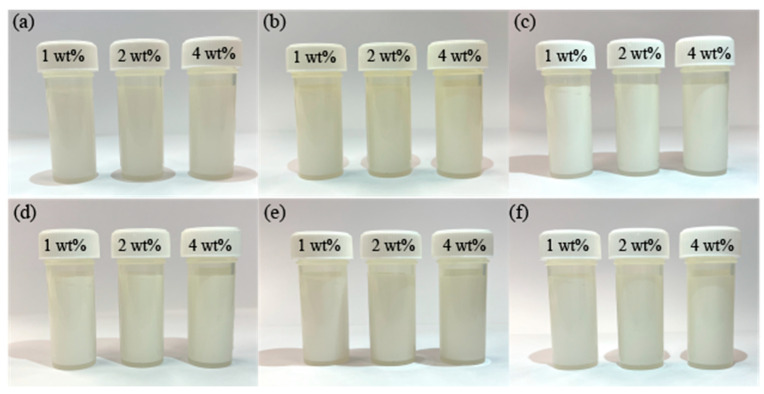
Sedimentation of TiO_2_ NPs dispersed in different glycerol-based nanolubricants at a settling time of (**a**) 0 h; (**b**) 24 h; (**c**) 48 h; (**d**) 72 h; (**e**) 96 h; and (**f**) 120 h.

**Figure 7 materials-16-02196-f007:**
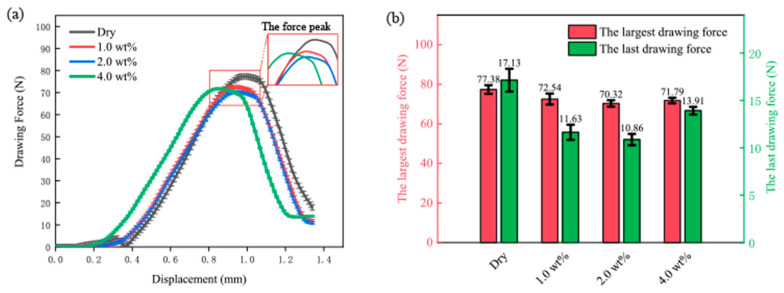
(**a**) Drawing force-displacement curve under different lubrication conditions; and (**b**) the largest and last drawing force under different lubrication conditions.

**Figure 8 materials-16-02196-f008:**
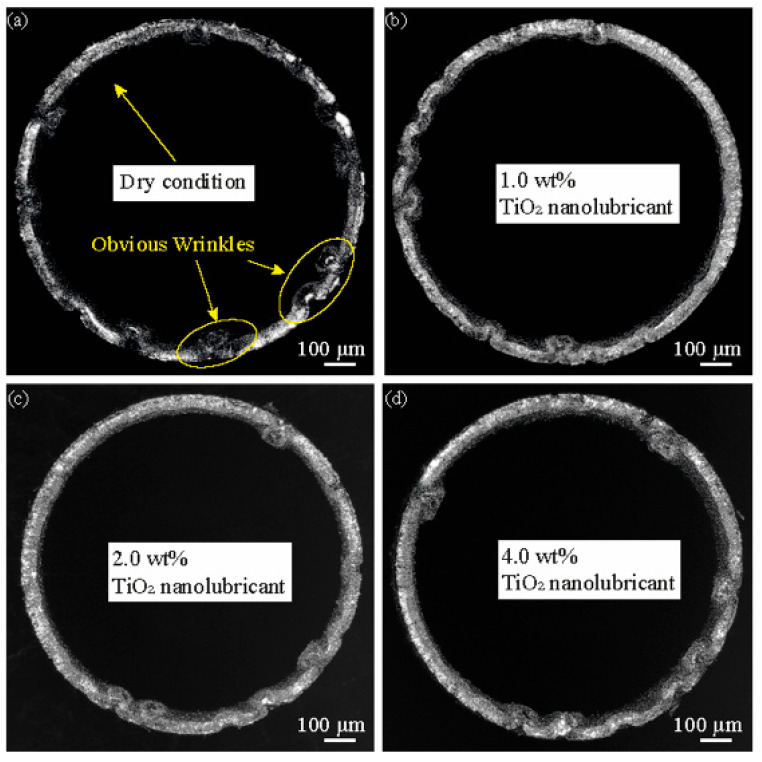
Mouth view of drawn cups from different lubrication conditions: (**a**) dry; (**b**) 1 wt% TiO_2_ lubrication; (**c**) 2 wt% TiO_2_ lubrication; and (**d**) 4 wt% TiO_2_ lubrication.

**Figure 9 materials-16-02196-f009:**
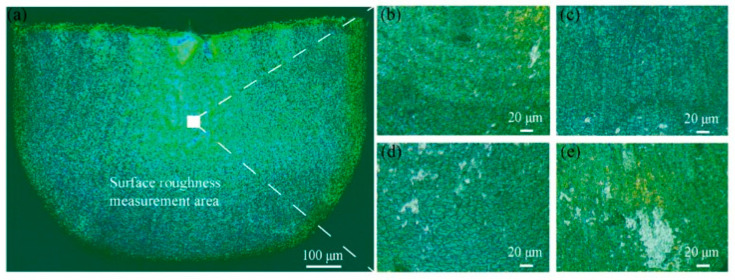
(**a**) The selected area on the cup for surface roughness measurement; and the surface morphology of the cups from (**b**) dry; (**c**) 1 wt% TiO_2_ lubrication; (**d**) 2 wt% TiO_2_ lubrication; and (**e**) 4 wt% TiO_2_ lubrication.

**Figure 10 materials-16-02196-f010:**
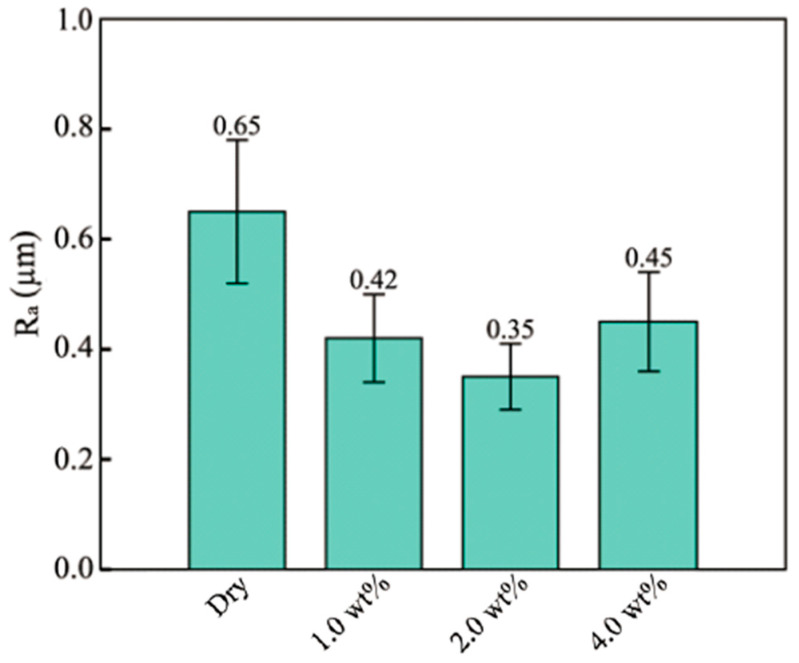
The average *R_a_* value of micro cup surface under different lubrications.

**Figure 11 materials-16-02196-f011:**
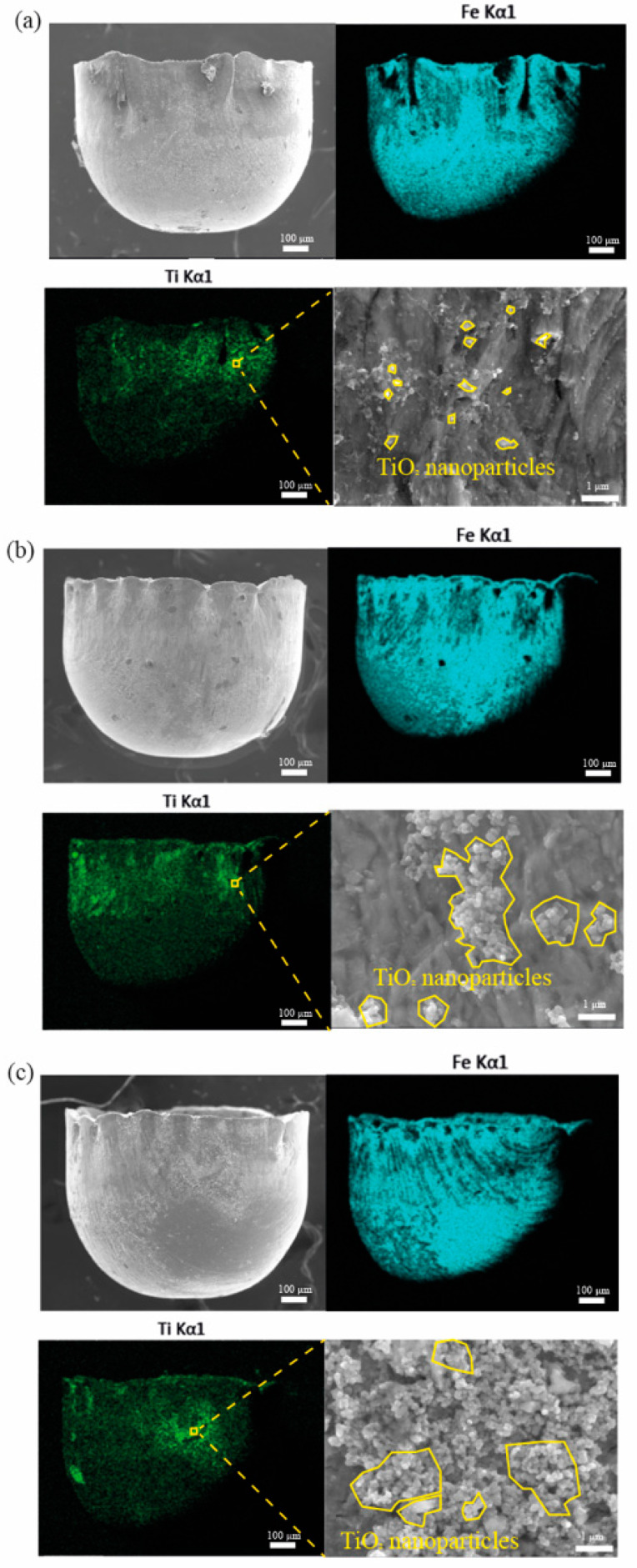
SEM images and EDS mappings of the micro cups surfaces lubricated by (**a**) 1 wt% TiO_2_ nanolubricant, (**b**) 2 wt% TiO_2_ nanolubricant, and (**c**) 4 wt% TiO_2_ nanolubricant.

**Figure 12 materials-16-02196-f012:**
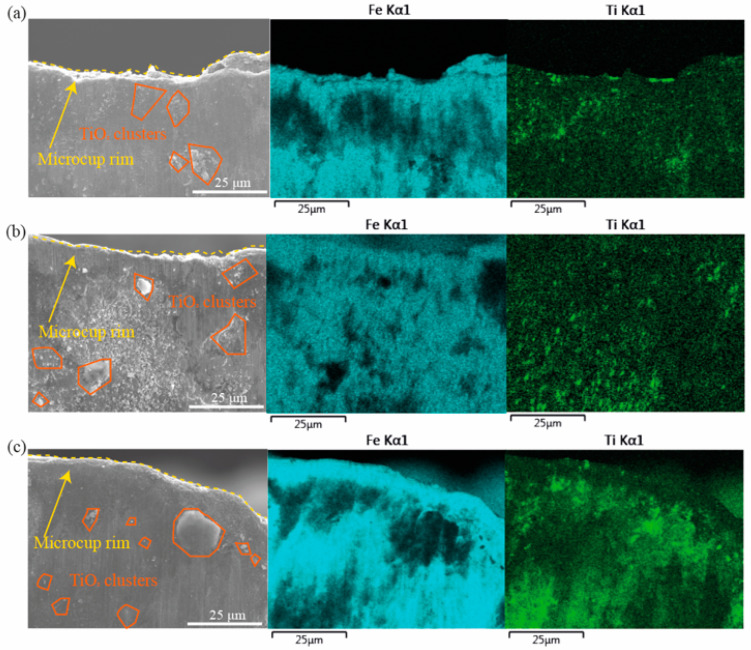
SEM images and EDS mappings of the micro cups mouth lubricated by (**a**) 1 wt% TiO_2_ nanolubricant, (**b**) 2 wt% TiO_2_ nanolubricant, and (**c**) 4 wt% TiO_2_ nanolubricant.

**Figure 13 materials-16-02196-f013:**
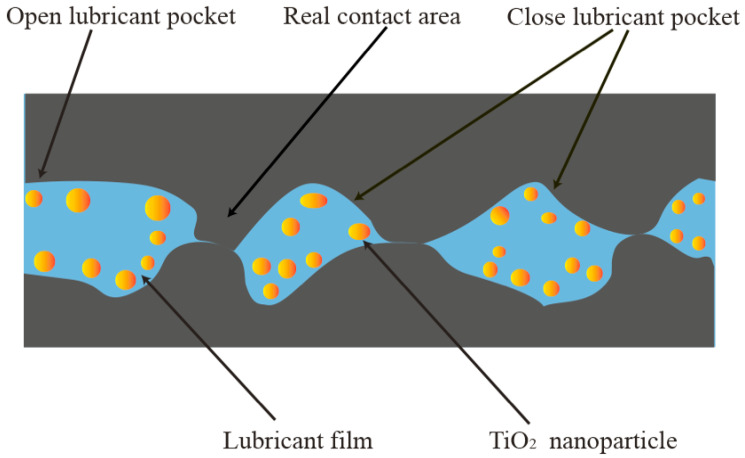
The schematic of the lubrication mechanism of TiO_2_ nanolubricant.

**Table 1 materials-16-02196-t001:** Chemical compositions of SUS301 (wt%).

C	Si	Cr	Mn	Ni	N	P	S	Fe
0.15	0.75	16.00–18.00	2.00	6.00–8.00	0.10	0.045	0.030	Balance

**Table 2 materials-16-02196-t002:** Parameters of the die set and blank.

Punch Diameter (mm)	Die Diameter(mm)	Radius of Punch Fillet (mm)	Radius of Die Fillet (mm)	Initial Blank Diameter (mm)
0.8	0.975	0.3	0.3	1.6

**Table 3 materials-16-02196-t003:** Various compositions of lubricants.

Lubricant Category	Description
1 wt% TiO_2_	1 wt% TiO_2_ + 0.4 wt% PEI + 80 wt% glycerol+ balance deionised water
2 wt% TiO_2_	2 wt% TiO_2_ + 0.4 wt% PEI + 80 wt% glycerol+ balance deionised water
4 wt% TiO_2_	4 wt% TiO_2_ + 0.4 wt% PEI + 80 wt% glycerol+ balance deionised water

**Table 4 materials-16-02196-t004:** Lubricant viscosity at room temperature.

Nanolubricants	Dynamic Viscosity (Pa∙s)
Glycerol solution	0.723 ± 0.03
1 wt% TiO_2_	0.749 ± 0.04
2 wt% TiO_2_	0.854 ± 0.03
4 wt% TiO_2_	0.872 ± 0.05

## Data Availability

Data from this study is available upon request.

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
