# Peer review of "Analysis of TiO2 Nanolubricant Influence in Micro Deep Drawing of Stainless Steel SUS301"

_materials, 2023, doi:10.3390/ma16062196_

Round 1

Reviewer 1 Report

Referee Report on “Analysis of TiO2 Nanolubricant Influence in Micro Deep 2 Drawing of SUS301

This is, of course, a work that could be recommended for publication, but only after some of the improvements formulated below.

1.       The title of the article must mention that it is steel.

2.       Titanium oxide has several modifications and each of them works in its own way. In addition, these effects depend on the size of the nanoparticles. This remains outside the introduction and is slightly misunderstood. https://scholar.google.lv/scholar?hl=ru&as_sdt=0%2C5&q=TiO2+steel&btnG=

3.       Line 42 -43. A quite important article that should be mentioned here: Tsebriienko, T., & Popov, A. I. (2021). Effect of poly (titanium oxide) on the viscoelastic and thermophysical properties of interpenetrating polymer networks. Crystals11(7), 794.

4.       Line 60. This sentence needs supporting references.

5.       Table 3. These data need error bars and corresponding explanation in the text.

6.       Fig.4. How stable are these pictures over time? Are there aging effects?

7.       Can you clearly formulate in your conclusions what new data on these materials were obtained in this work.

In principle, the article is interesting and can be recommended for publication after due consideration of the above comments.

Reviewer 2 Report

The submitted paper reports an interesting study of the effect of TiO2 nanolubricant on the MDD process. The topic is of interest, but the manuscript has to be modified in some of its parts to be reconsidered for the final acceptance.

More in details, the authors are asked to better clarify the following list of aspects.

1. What was the reason to anneal the SUS foils? Was the combination of time and temperature (980°C for 2 mins) taken from literature? If so, the reference work should be cited.

2. The authors should better explain the need of the microtensile tests: there's a lack of information regarding: (i) the specimen's dimension, (ii) the reference standard, (iii) the adopted machine for the tensile tests, (iv) the flow curve, (v) the number of replications.

3. In section 3.2, what kind of information could be retrieved from the last drawing force? Moreover, by looking at the load curve in Figure 5a, why is the 4% curve characterized by an earlier peak value?

4. In the description of the MDD tests, there's no mention regarding the applied BHF. Did the author take into account the influence that the BHF may have had on the wrinkling tendency (in combination with the TiO2 nanoparticles percentage)?

5. Did the authors take into account if the punch speed (that may have an influence on the fluid viscosity) could affect the outcome of the tests?

6. Did the author also evaluate how the different percentage could have an influence on the final thickness distribution of MDD cups?

Reviewer 3 Report

To improve the manuscript, please provide additional information and perform the following changes:

1) The weight contents of TiO2 NPs in the lubricant should be reported without one decimal of “0” since the values are integers.

2) On page 2, section 2. Materials and Methods should be specified the supplier and the other sizes (length x width) of SUS301 foils with a thickness of 40 ± 2 μm.

3) On page 2, lines 70-72 should be specified the type of equipment, along with the size of the annealed SUS301 foil samples and the measurement conditions used in micro tensile tests In the comment ”The average yield strength of the annealed foil is 387 MPa.” should be added the standard deviation value beside the value of the average yield strength.

4) The size of the SUS301 foil samples used in the MDD tests, and the volume of the nanolubricant dropped into the die cavity before the MDD tests (page 3, lines 94-95).

5) On page 3, section 2. Materials and Methods should be specified the measurement conditions used in the XRD analysis, TEM, and EDS, and to determine the viscosities of the lubricants.

6) On page 3, section 2. Materials and Methods, lines 86-87, referring to the comment “Different weight concentrations (1.0 wt%, 2.0 wt% and 4.0 wt%) of glycerol-based TiO2 NP lubricants were prepared to be utilised in the MDD” should be clarified the producer of the constitutive components of the glycerol-based TiO2 NP lubricants. Some basic properties of the TiO2 NPs (purity, shape, and size) should also be specified. The use, type, and content of a dispersing agent should be clarified in the preparation of the glycerol-based TiO2 NP lubricants.

7) The hydrodynamic diameter, grain size distribution, and stability (zeta potential) of glycerol-based TiO2 NP lubricants should be performed by dynamic light scattering (DLS) and electrophoretic light scattering (ELS) measurements.

8) On page 4, in subsection 3.1. Characterisation of TiO2 nanolubricant, the phase identification in the XRD analysis should be supported by the number of PDF reference cards for TiO2 rutile and TiO2 anatase issued by the International Centre for Diffraction Data (ICDD).

9) On page 5, Table 3, the values of the dynamic viscosity of the glycerol-based TiO2 NP lubricants should be reported as mean ± standard deviation. The dynamic viscosity of the TiO2-free glycerol-based lubricants should also be clarified.

10) In the comments related to the surface quality of the micro cup should be reported the values of the surface roughness Ra as mean ± standard deviation to be correlated with the values (mean and error bars) shown in Figure 8. The surface roughness Ra of the SUS301 foil samples before performing the MDD tests should also be clarified.

11) In Figures 9 and 10 should be added the EDS spectra. The lack of the oxygen (O) element is unclear in the actual results, since TiO2 clusters were indicated in some images. The elemental content should also be specified.

12) On page 10, lines 237-244, should be clarified how were achieved and monitored the data related to the contact pressure and coefficient of friction. Some words should be corrected (on line 238, “berar” with “bear”, and on line 239, “The contact pressue has a great imapct…” with “The contact pressure has a great impact…”).

Round 2

Reviewer 1 Report

The authors have strongly improved their original manuscript, which now can be recommended for publication

Author Response

Comment: The authors have strongly improved their original manuscript, which now can be recommended for publication

Response: Thank you for your review. I am pleased to hear that the revisions have significantly strengthened the manuscript

Reviewer 2 Report

The comments have been satisfactorily addressed

Author Response

Comment: The comments have been satisfactorily addressed.

Response: Thank you for your review. I am glad to hear that the comments have been addressed to your satisfaction.

Reviewer 3 Report

The revision is in general satisfactory. However, several amendments and clarifications are necessary:

(1) On page 2, line 79, in "5000 x 5000 µm (length x width)" should be written as "5000 µm x 5000 µm (length x width)";

(2) On page 2, line 87, the year of standard ISO 12086-2:1955 is wrong. According to the information on https://www.iso.org/standard/37663.html the former year of the standard was 1995 then 2006, and the standard in force is ISO 12086-2:2017 “Plastics — Fluoropolymer dispersions and moulding and extrusion materials — Part 2: Preparation of test specimens and determination of properties”. However, it is unclear how the standard that describes the preparation of test specimens and provides test methods to define characteristics of thermoplastic fluoropolymer resins is suitable to the test specimens made of metal materials such as stainless steel SUS301 foils used in this study. Please clarify this issue for applying the same standard on dissimilar materials.

(3) On page 4, the chemical compositions of the glycerol-based TiO2 NP lubricants is unclear, since the final volume of the prepared lubricants should be mentioned. The formulation “First, a specific amount of pure TiO2 NPs (P25 sourced from Sigma-Aldrich™ with an approximate diameter of 20 nm) were weighed and added to balanced deionised water” is unclear on the initial content of TiO2 NP and volume of the aqueous suspension. Referring to “The mixture was then stirred mechanically” should be provided the stirring speed and stirring time.

(4) On page 4, lines 121 - 122, in the phrase “The XRD analysis was then conducted using a Philips PW1730 conventional diffractometer and Cu-Kα radiation at room temperature.” should be specified the measurement conditions (2Theta range, step size, and time per step) used in the XRD analysis.

(5) On page 4, lines 142 – 143, in the comment “The phases of these NPs can be determined as a typical P25 TiO2, which contains 25% rutile and 75% anatase, referring to the XRD standard atlas [47].” should be specified the number of the PDF reference cards for TiO2 rutile and TiO2 anatase. The authors stated in their answer to the reviewer comments that “included the relevant PDF reference cards for TiO2 rutile and TiO2 anatase from the International Centre for Diffraction Data (ICDD) in the XRD analysis section (Page. 4, Line 140, highlighted with blue)” but this information was not provided.

(6) On page 5, lines 151 – 155, referring to the comments “To address concerns about stability, extensive observational tests were conducted on the TiO2 nanolubricant after ultrasonic treatment at intervals of 1 day, 10 days, and 30 days. The results indicate that the nanoparticles maintain their size and dispersion throughout the observation period, under various conditions, demonstrating the long-term stability of the system.”, it is still unclear the type of the observational tests or characterization methods (adequate methods should be dynamic light scattering (DLS) and electrophoretic light scattering (ELS)). The time of ultrasonic treatment is not specified and why the ultrasonication step was necessary if the suspensions were stable. The comment on “under various conditions” is unclear on the type and values of the variable conditions. The results in the initial state and after 1 day, 10 days, and 30 days, or at least in the initial and after 30 days, are not presented. The particle size (hydrodynamic diameter), grain size distribution, and long-term stability of the suspensions should be proved by histograms showing particle size distributions (determined by DLS analysis) and zeta potential (determined by ELS analysis). The results shown in Figure 5. TEM images of TiO2 NPs dried from different weight concentrations lubricants are insufficient to prove the raised questions on the size and stability of the TiO2 NPs suspended in a content of 1 wt.%, 2 wt.%, and 4 wt.% in the glycerol-based lubricant containing deionised water (page 4, line 112) and 0.4 wt.% polyethyleneimine (PEI) (page 4, line 113).

(7)  The TEM image from Figure 5 (b) for TiO2 NPs dried from 2 wt.% TiO2 NPs lubricant is identical with the one shown in another study in Fig. 4. TEM images of TiO2 nanoparticles dried from different mass concentrations of water-based lubricants: (c) 4.0 wt.% with the chemical composition of the lubricant: 4.0 wt.% TiO2+0.04 wt. % PEI + 10.0 wt.% glycerol + balance water (please see the article published by Hui Wu, Jingwei Zhao, Wenzhen Xia, Xiawei Cheng, Anshun He, Jung Ho Yun, Lianzhou Wang, Han Huang, Sihai Jiao, Li Huang, Suoquan Zhang, Zhengyi Jiang, Analysis of TiO2 nano-additive water-based lubricants in hot rolling of microalloyed steel, Journal of Manufacturing Processes, Volume 27, 2017, Pages 26-36, ISSN 1526-6125, https://doi.org/10.1016/j.jmapro.2017.03.011, https://www.sciencedirect.com/science/article/pii/S1526612517300683), provided as reference [38] in this study. This issue raise questions on the credibility of the TEM results. It is unethical to publish the same results on different studies, especially the chemical composition of the glycerol-based TiO2 NP lubricants was not identical in both studies. Therefore, the results on DLS and ELS are necessary to support the credibility of the study.

(8) On page 4, lines 118 – 120, in the comment “In order to determine the phase of TiO2 NPs, a sample of the nanolubricant was prepared for power X-ray diffraction (XRD) analysis. This was achieved by spreading a thin film of the lubricant onto a glass substrate”, it is unclear which one of the 1 wt.% TiO2, 2 wt.% TiO2 NPs and 4 wt.% TiO2 lubricant was used. In addition, the diffraction pattern is similar with the one shown in Fig. 3. XRD pattern of TiO2 nanoparticles of the above-mentioned study published in the Journal of Manufacturing Processes. In this case, the authors could refer that the XRD diffraction pattern was published elsewhere (in reference [38]).

Round 3

Reviewer 3 Report

The revision is in general satisfactory. However, the following amendments should be performed:

(1) On page 2, line 79, in the comment “SUS301 foils with a 5 x 5 mm (length x width) and thickness of 40 ± 2 µm…” should be added the measurement unit of the length because in this form only for the width is shown.

(2) In Figure 1, in the Oy axis, the measurement unit of the True Stress should be written as MPa instead of Mpa.

(3) Referring to Table 1. Chemical compositions of SUS301, the measurement unit is missing. Probably the chemical composition is in wt.%, but it should be specified in the manuscript. The same observation is for the comment on page 5, lines 154-157 "The phases of these NPs can be determined as a typical P25 TiO2, which contains 25% rutile and 75% anatase, referring to the XRD standard atlas as (JCPDS Nos. 21–1272 for Anatase-type and 21–1276 for Rutile-type)" to clarify if the content of anatase and rutile is expressed in wt.%.
